# Design of SWB MIMO Antenna with Extremely Wideband Isolation

**Habib Ullah [1] , Saeed Ur Rahman [1] , Qunsheng Cao [1],\*, Ijaz Khan [2] and Hamid Ullah [3]**

[1]  College of Electronic and Information Engineering, Nanjing University of Aeronautics and Astronautics (NUAA), Nanjing 210016, China; habibullah@nuaa.edu.cn (H.U.); saeed@nuaa.edu.cn (S.U.R.)
[2]  Herbin Institute of Technology, College of Information and Communication Engineering, Herbin 150001, China; khanijaz438@yahoo.com
[3]  School of Information Science and Technology, Southwest Jiaotong University, Chengdu 611756, China; hamid.jan54@yahoo.com
\*  Correspondence: qunsheng@nuaa.edu.cn

**Abstract:** This paper presents a compact planar multiple input multiple output (MIMO) antenna for super wide band (SWB) applications. The presented MIMO antenna comprises two identical patches on the same substrate. Dimensions of the MIMO antenna are $0.17\lambda \times 0.20\lambda \times 0.006\lambda$ mm$^3$, with respect to the lowest resonance of 1.30 GHz. The SWB antenna was manufactured using F4B substrate having a dielectric constant of 2.65 that provides a percent impedance bandwidth and bandwidth ratio of 187% and 30.76:1, respectively. The mutual coupling between the antenna elements is suppressed by placing a T-shaped corrugated strip in the mid of two antenna elements. The proposed MIMO antenna exhibits maximum diversity gain of 10 dB, low mutual coupling ($<-20$ dB), low envelope correlation coefficient (ECC < 0.02), efficiency >80%, and low reflection coefficient ($<-10$ dB) in the SWB frequency range (1.30 GH–40 GHz). The presented antenna is a good candidate for SWB applications. The designed antenna has been experimentally validated, and the simulated results were also verified.

**Keywords:** super wideband; MIMO antenna; wideband isolation

## 1. Introduction

In advanced wireless communications, extremely wideband antennas have gained rising popularity because of their low manufacturing cost, high data rate, convenient design, and provision of wide-range of resonance frequencies. The ultra-wideband (UWB) and super wideband (SWB) planar monopole antennas are good candidates for both long and short-range communication. The bandwidth (BW) ratio of SWB antennas is greater than 10:1. Multiple wideband antennas have been designed for UWB [1–5] and SWB [6–15] applications. The large dimensions/size of such wideband antennas is a major challenge. In [16], the dimensions of the proposed wideband antenna that resonates for a wideband range (2.67 GHz–10 GHz) are $0.56\lambda \times 0.56\lambda$ mm$^2$. Similarly in [3], a coplanar waveguide (CPW)-fed antenna has been designed for UWB applications with dimensions of $0.43\lambda \times 0.84\lambda$ mm$^2$. Mostly the previously designed [1–8] wideband antennas have large dimensions, low % BW, and high lowest resonance frequency. For example, the dimensions of the SWB antenna designed in [6] is $2.0\lambda \times 2.0\lambda$ mm$^2$, with a 5:1 BW ratio and a %BW of 164%.

With the number of users of wireless communications on the rise, the data rate requirements are dynamically increasing. For example, presently WLANs offer data rates up to 54.0 Mbps with an envisaged increase up to 600 Mbps in the near future. The audio/video streams, such as those in high-definition television, require 1 Gbps transmission. To improve the channel capacity, as in 5 G

technology, multiple-input multiple-output (MIMO) antenna is a good choice that can offer massive data transmission rates [17]. MIMO is a multiplexing technology that utilizes multiple antennas at both receiver and transmitter ends resulting in improved communication performance at both ends [18]. On the other hand, UWB/SWB is another technology that can provide massive data transmission rates by utilizing a very wide frequency band. Extensive research has been conducted in UWB and SWB antennas [1–8]. More advanced technology needs to be used when the required data rate reaches 1 Gigabyte per second. A combination of MIMO and extremely wideband technology such as UWB and SWB may offer a viable solution.

In the reported literature, many MIMO antennas for UWB applications have been designed [19–22]. Mostly, the previously designed MIMO antennas operate within the UWB band (3.1 GHz–10.6 GHz) and they possess large electrical dimensions, low %BW, and high lowest resonance frequency. For example, the designed MIMO antenna in [21] operates within UWB frequency band and has electrical dimensions of $0.41\lambda \times 0.25\lambda$ mm$^2$. The electrical dimensions of the UWB MIMO antenna in [23] are $0.41\lambda \times 0.47\lambda$ mm$^2$ and in [24–26] the dimensions are $0.25\lambda \times 0.33\lambda$ mm$^2$, $0.96\lambda \times 1.11\lambda$ mm$^2$, and $0.41\lambda \times 0.33\lambda$ mm$^2$ respectively. A number of challenges are being faced in the design of MIMO antennas. One of the challenges is to minimize the number of elements for UWB system and to keep the size of the antenna elements compact. The other one is to enhance the isolation between the two antenna elements in the operating band. Enhancing isolation may affect the wideband impedance matching but low isolation between the antenna elements decreases the data transfer rate and efficiency of the MIMO system. Several methods have been employed to increase the isolation or to reduce the mutual coupling among MIMO antenna elements. Isolation has been enhanced by placing the radiators of two monopole antenna elements in opposite directions, by introducing of n-section slits below the feed line [24], using carbon black film [26], etching a slot at the center of the ground [27], introducing F-shaped stubs in the shared ground plane [21], utilizing the defected ground structures [28] and by placing the antenna elements perpendicularly to each other [29]. All these techniques have been employed to reduce coupling in the range 3.1 GHz–10.6 GHz. Further, mutual coupling reduced by metamaterial in [30–39] and [40–43]. Most of these antennas possess large dimensions, have low isolation enhancement and low %BW and gain. For example, in [23] the MIMO antenna possesses large physical dimensions of $58.6 \times 46$ mm$^2$ with a mutual coupling of $<-13$ dB, in [25] the mutual coupling is $<-17$ dB whereas the size of the MIMO antenna is $50 \times 80$ mm$^2$ and the isolation enhancement of $40 \times 68$ mm$^2$ MIMO antenna designed in [22] is less than $-15$ dB.

In this paper, we have designed a SWB MIMO antenna that provides an extremely wideband impedance matching from 1.30 GHz to 40 GHz. The designed wideband MIMO antenna exhibits percent impedance BW and BW ratio of 187% and 30.76:1 respectively. The maximum peak gain of the MIMO antenna linearly increases and reaches 10.9 dB. The average peak gain within resonating band is 6.7 dB and diversity gain is about 10 dB. The tapered feed line technique has been employed to enhance the impedance matching. The T-shape corrugated strip is placed among antenna elements to reduce mutual coupling. The mutual coupling is $<-20$ dB within the resonating band (1.30 GHz–40 GHz), and the envelope correlation coefficient (ECC) is less than 0.02. To best of our knowledge very few highly isolated MIMO antennas with lowest resonance frequency of 1.3 GHz and highest resonance frequency of 40 GHz has been designed for SWB application.

## 2. SWB Antenna Design

In this section, a planar monopole antenna is designed for SWB application. The schematic diagram of single SWB antenna is shown in Figure 1a. The designed planar monopole antenna possess a half circular disc shape radiator and rectangular shape ground. The planar monopole antenna is etched on F4B substrate having 1.5 mm thickness. To achieve optimum matching the designed antenna is fed with tapered micro-strip feed line as shown in Figure 1a. The tapered feed line enhance the impedance bandwidth [9]. The single element has wide impedance matching from 1.7 GHz to more than 40 GHz. To verify the designed SWB antenna, the E-plane co- and cross polar radiation

pattern is depicted in Figure 2 at (a) 5 GHz, (b) 10 GHz, (c) 20 GHz, and at (d) 30 GHz. At lower resonance frequencies co-polar pattern is greater than cross-polar with Omni-directional behavior such as at 5 GHz and 10 GHz. At higher resonance frequencies the Omni-directional pattern becomes distorted and nulls are created. Because of this distortion, at some angles, the co-polar pattern is less than cross-polar pattern. It happen in mostly wideband antenna because of the excitation of hybrid modes at higher frequencies. However, the proposed monopole antenna maintain nearly stable radiation pattern.

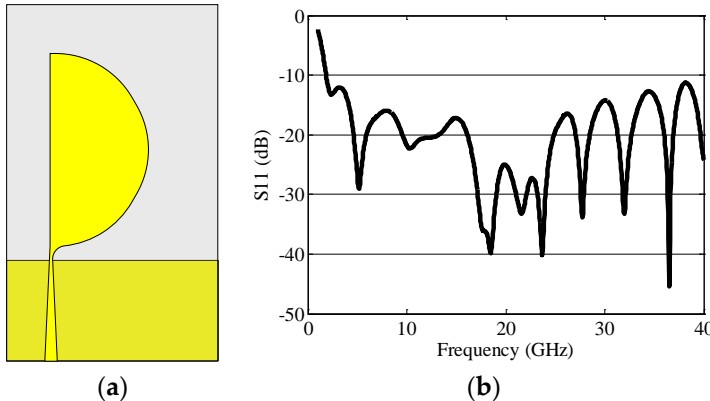

(**a**)  (**b**)

**Figure 1.** (**a**) The schematic design of single super wide band (SWB) antenna element and (**b**) S11.

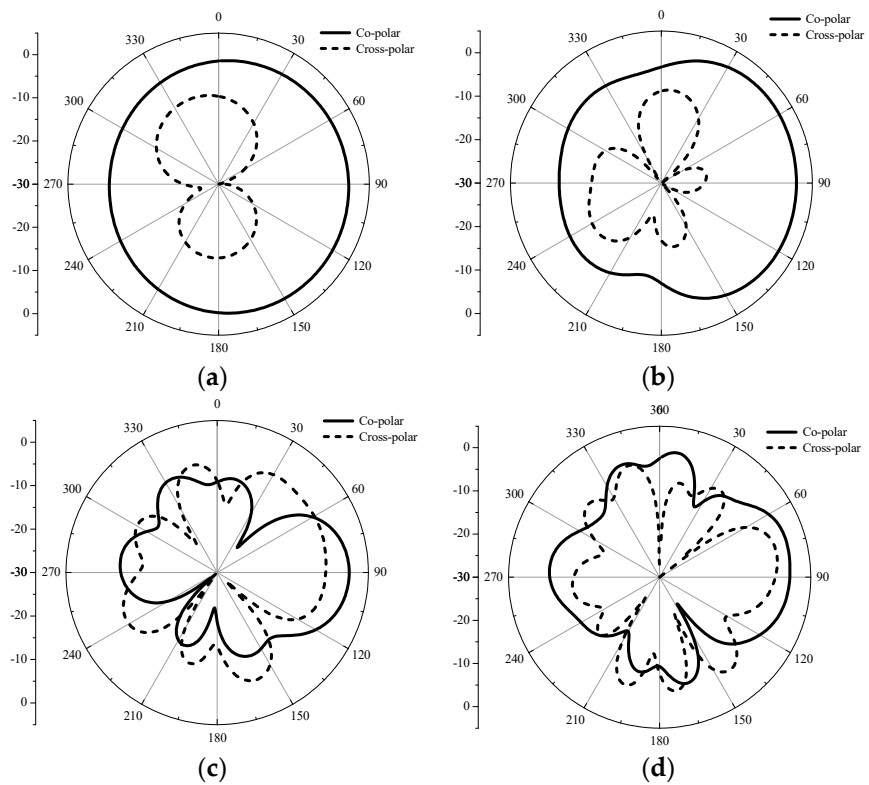

(**a**)  (**b**)

(**c**)  (**d**)

**Figure 2.** The co- and cross polar pattern at (**a**) 5 GHz, (**b**) 10 GHz, (**c**) 20 GHz, and (**d**) 30 GHz.

## 3. Results MIMO Antenna Design

Two monopole antenna elements with circular shape radiator are chosen as shown as design I in Figure 3. The S-parameters for design I are depicted in Figure 4a,b. The design I shown in Figure 3 exhibits impedance mismatching for many frequencies, so the circular shape radiator was modified to a semi-circular shape radiator, shown as design II in Figure 3. The simulated impedance

matching for semicircular shape radiator monopole antenna is shown in Figure 4a. The design II has comparatively better impedance matching than design I, where the coupling among antenna elements is very high as shown in Figure 4b. As the tapered feed line provides wideband impedance matching [7], the rectangular micro-strip feed line is modified to tapered feed line.

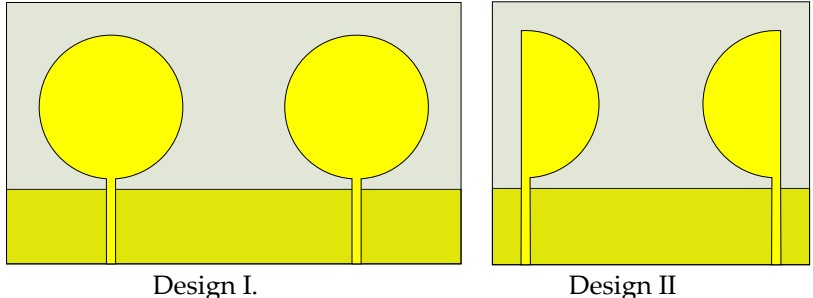

Design I.          Design II

**Figure 3.** The design procedure of the proposed SWB multiple input multiple output (MIMO) antenna.

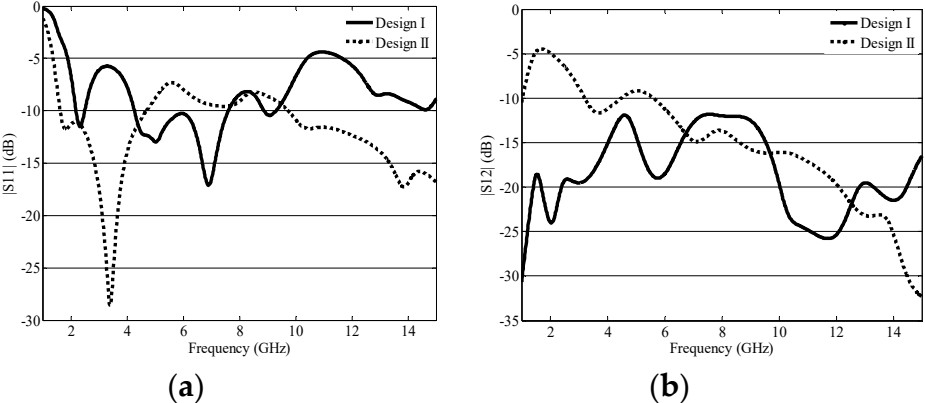

(**a**)          (**b**)

**Figure 4.** The simulated S-parameters (**a**) S11 and (**b**) S12.

The schematic diagram of the proposed design with tapered feed line is depicted in Figure 5a. The designed MIMO antenna has compact dimensions of $40 \times 47$ mm$^2$ (Ls × Ws), and the rectangular ground has a length Lg = 11.4 mm. The designed MIMO antenna's tapered 50 Ω feed has a width of 1.4 mm which linearly decreases to $w_t$ = 0.25 mm.

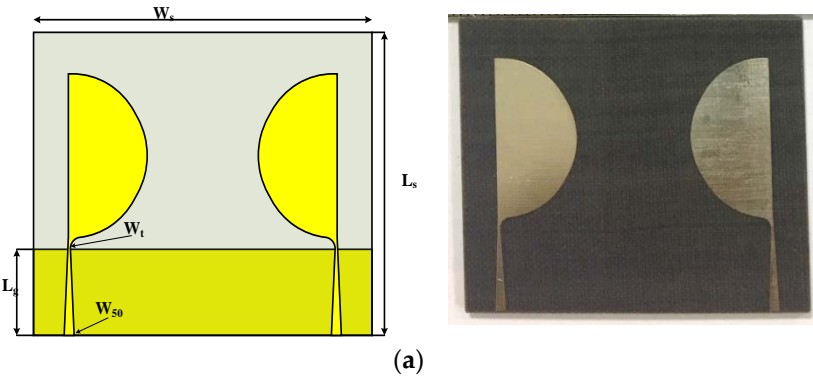

(**a**)

**Figure 5.** *Cont.*

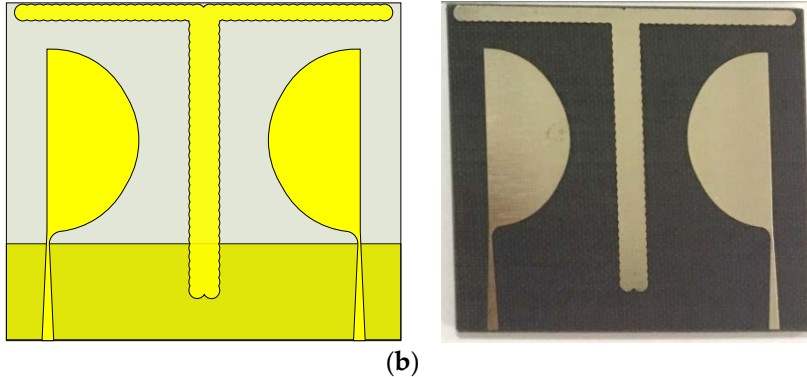

(**b**)

**Figure 5.** The schematic diagram (**a**) (**left side**) fabricated (**right side**) without t-shape corrugated strip and (**b**) (**left side**) fabricated (**right side**) with t-shape corrugated strip of the MIMO antenna.

Simulations were carried out using Microwave studio computer simulation technology (CST) software. Finally, schematic design shown in Figure 5a was fabricated on F4B substrate with a permittivity of 2.65 and a thickness of 1.5 mm. The measurements were made by using a two-port Agilent N5245A PNA-X network analyzer. The measured and simulated scattering parameter of the manufactured antenna are given in Figure 6a,b. As can be seen, the measured scattering parameters validate the simulated scattering parameters. The impedance matching S11 is given in Figure 6a. From Figure 6a it is clear that S11 is less than −10 dB for the 1.3 GHz to 40 GHz. In addition, the measured and simulated S12/S21 is less than −10 dB; however, at some frequencies it increases, signifying that the isolation is very low at lower frequencies. Very little discrepancy is observed between the measured and simulated results owing either to the fabrication tolerance/manufacture accuracy or to the SMA connector.

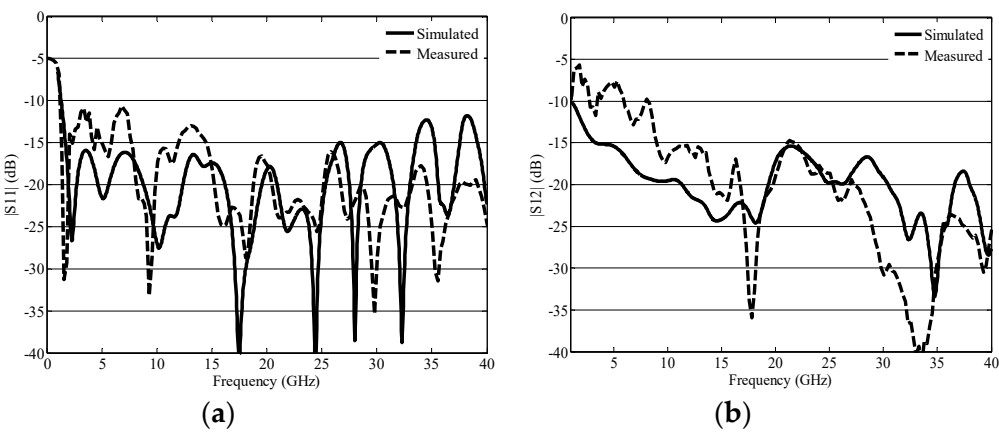

(**a**)        (**b**)

**Figure 6.** The simulated and measured s-parameters (**a**) s11 and (**b**) s12.

## 4. Wideband Isolated MIMO Antenna

The decoupling setup provides negative coupling, which cancels out the coupling caused by the adjacent excited antenna element, thus reducing the mutual coupling among the antenna elements. The mutual coupling is mostly caused by surface waves and space waves. Power dissipation takes place among the antenna elements because of parasitic resistance whose occurrence cannot be ignored. The decoupled structure complicates the structure of antenna. However, to increase the isolation, a number of MIMO antennas employ the decoupled structure. Parasitic elements are mostly used to enhance isolation as they produce an opposing coupled field that diminishes the coupling between the antenna elements, thus eliminating the RF current in the adjacent antenna element. The BW, the decoupling range, and the surface current coupling can be controlled by designing an appropriate

parasitic element. In this work, a corrugated T-shape strip parasitic element is selected to suppress the mutual coupling among antenna elements at lower frequencies as well as at higher frequencies. By placing the corrugated T-shape parasitic element, the impedance matching is not affected, as it is less than −10 dB in the desired frequency range (1.3 GHz–40 GHz).

### 4.1. Return Loss, Mutual Coupling, and Current Distribution

The measured and simulated S-parameter is given in Figure 7a,b. The measured and simulated S-parameter S11 is less than −10 dB for 1.3 GHz to 40 GHz as given in Figure 7a. The measured and simulated S-parameter S21 is given in Figure 7b. It can be seen that the corrugated T-shape parasitic element provides wideband isolation of S21 < −20 dB.

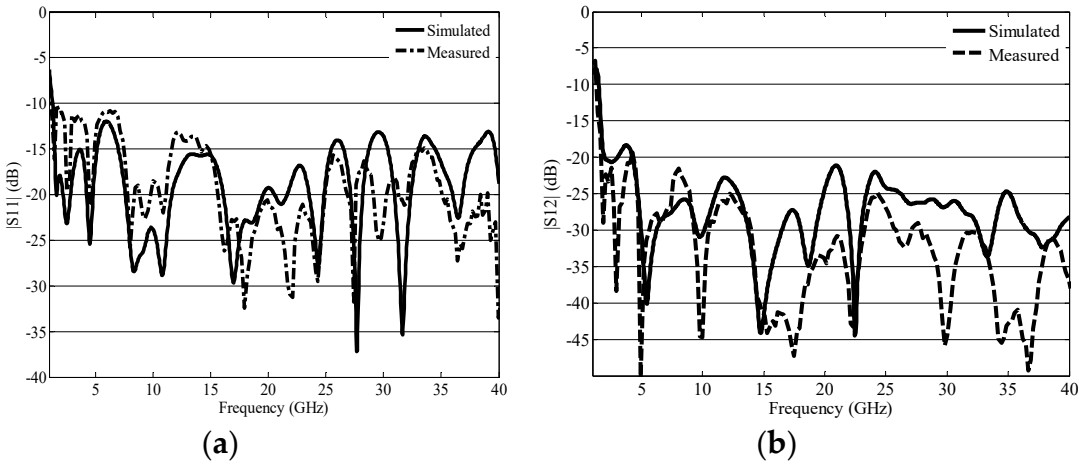

**Figure 7.** Simulated and measured S-parameters (**a**) S11 (**b**) S21 with t-shape corrugated parasitic element.

The coupled radiation leads the current on the adjacent antenna therefore the designed isolated structure receives the coupling fields and transforms them in surface current leading to an increase in isolation. To examine the effect of the isolated structure, the surface current distributions at 5, 10, and 15 GHz are shown in Figure 8a–c. The surface current distribution shows the isolation of MIMO antenna with loaded corrugated T–shape parasitic element. The surface current is mainly concentrated on the corrugated T–shape parasitic element which reduces the induced current on the neighboring antenna. In all cases, the left side monopole patch antenna was excited to see the impact of corrugated T–shape parasitic element, where the other element was terminated with a matched load. Bulleted lists look like this:

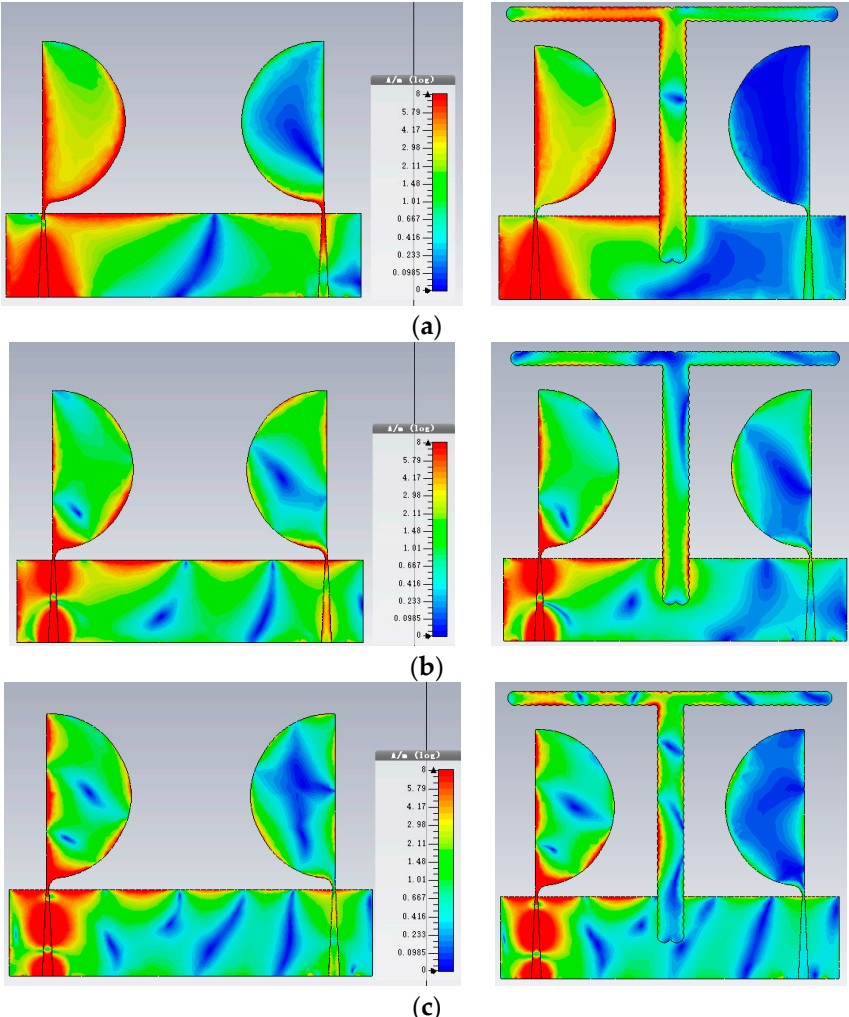

**Figure 8.** Surface current distribution without (**left side**) and with (**right side**) corrugated t-shape parasitic element at (**a**) at 5 GHz (**b**) 10 GHz and (**c**) 15 GHz.

*4.2. Radiation Patterns, Gain and ECC*

In order to confirm the suitability of the designed antenna for MIMO applications, it is important to realize a low ECC. The ECC is a measurement of how much the communication channels are isolated or correlated with each other. The following formula is used to calculate ECC.

$$ECC = \frac{\left|S_{11}^* S_{12} + S_{21}^* S_{22}\right|^2}{\left(1 - \left(|S_{11}|^2 + |S_{21}|^2\right)\right)\left(1 - \left(|S_{22}|^2 + |S_{12}|^2\right)\right)} \tag{1}$$

$$DG = 10 \sqrt{(1 - ECC^2)} \tag{2}$$

An ECC value of 0.5 has been set as a suitable value for diversity conditions. The ECC for the proposed MIMO antenna has been calculated from the scattering parameters using Equation (1) and a plot of ECC verses frequency is given in Figure 9a. The average ECC of the proposed MIMO antenna is 0.0073. Over the resonating band, the ECC is less than 0.02; however, the ECC slightly increases to 0.023 at 3.5 GHz. Diversity gain (DG) is another important parameter to measure the performance of the antenna. DG was calculated using Equation (2) and a plot of DG verses frequency is given in Figure 9b. From Figure 9b, it is clear that the DG is varying around 10 dB at all resonance frequencies. The peak gain with and without corrugated T–shape parasitic element is shown Figure 9c. The peak

gain of the MIMO antenna increases; however, the gain of the decoupled structure is less than that of the MIMO antenna with no decoupling structure. The efficiency is varying around about 90% as shown in Figure 9c.

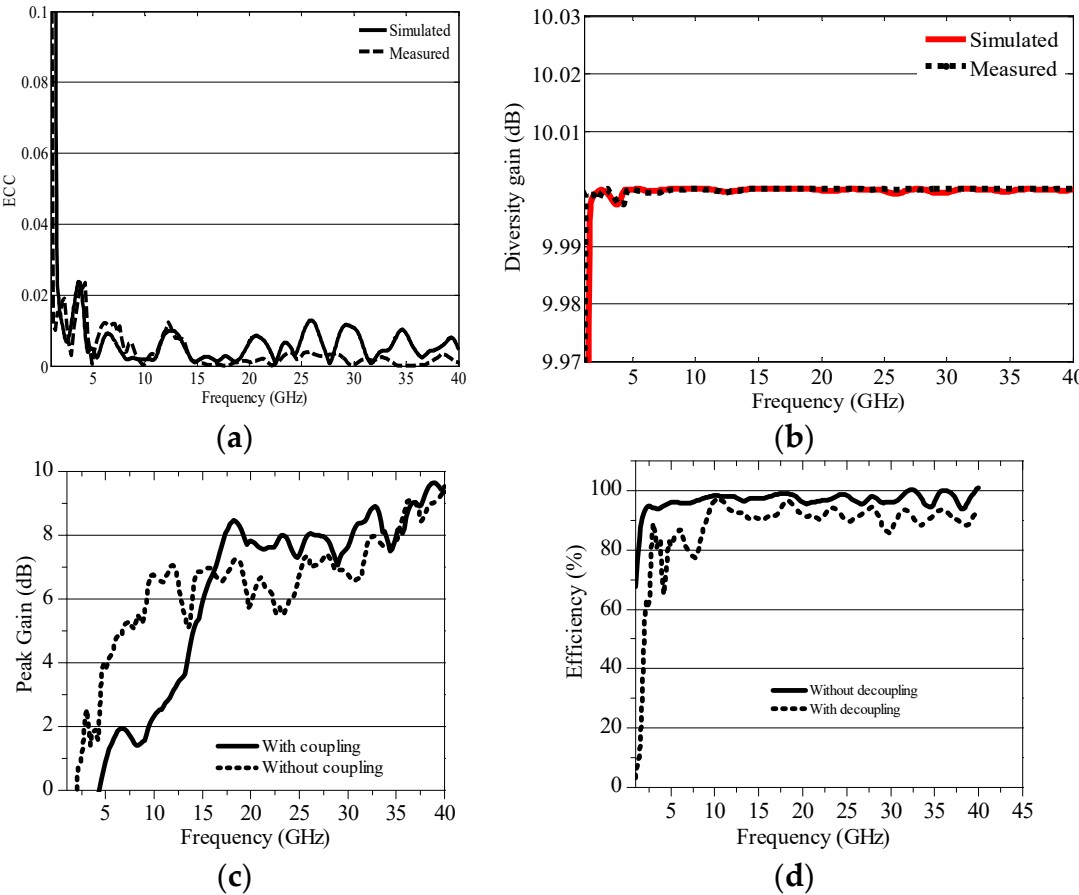

**Figure 9.** The plot of (**a**) envelope correlation coefficient (**b**) diversity gain (**c**) peak gain verses frequency and (**d**) efficiency.

Another important characteristic is the radiation pattern. The far-field radiation pattern in the H-plane and E-plane is depicted in Figure 10 at six different frequencies (a) 5 GHz, (b) 10 GHz, (c) 15 GHz, (d) 20 GHz, (e) 30 GHz, and (f) 40 GHz. The proposed antennas exhibits nearly omni-directional pattern in H-plane and bi-directional pattern in E-plane at lower frequencies from 1.3 GHz to 10 GHz. At higher frequencies, nulls are generated in both H-plane and E-plane. In addition, the proposed antenna has high co-polarization at lower frequency. While, at higher frequencies the co-polarization is lower than cross-polarization level, mainly because of (1) excitation of hybrid current distribution (2) horizontal component of the surface currents increases. Overall, the designed wide-band MIMO antenna exhibit nearly stable radiation pattern and matches the behavior of the planar monopole antenna.

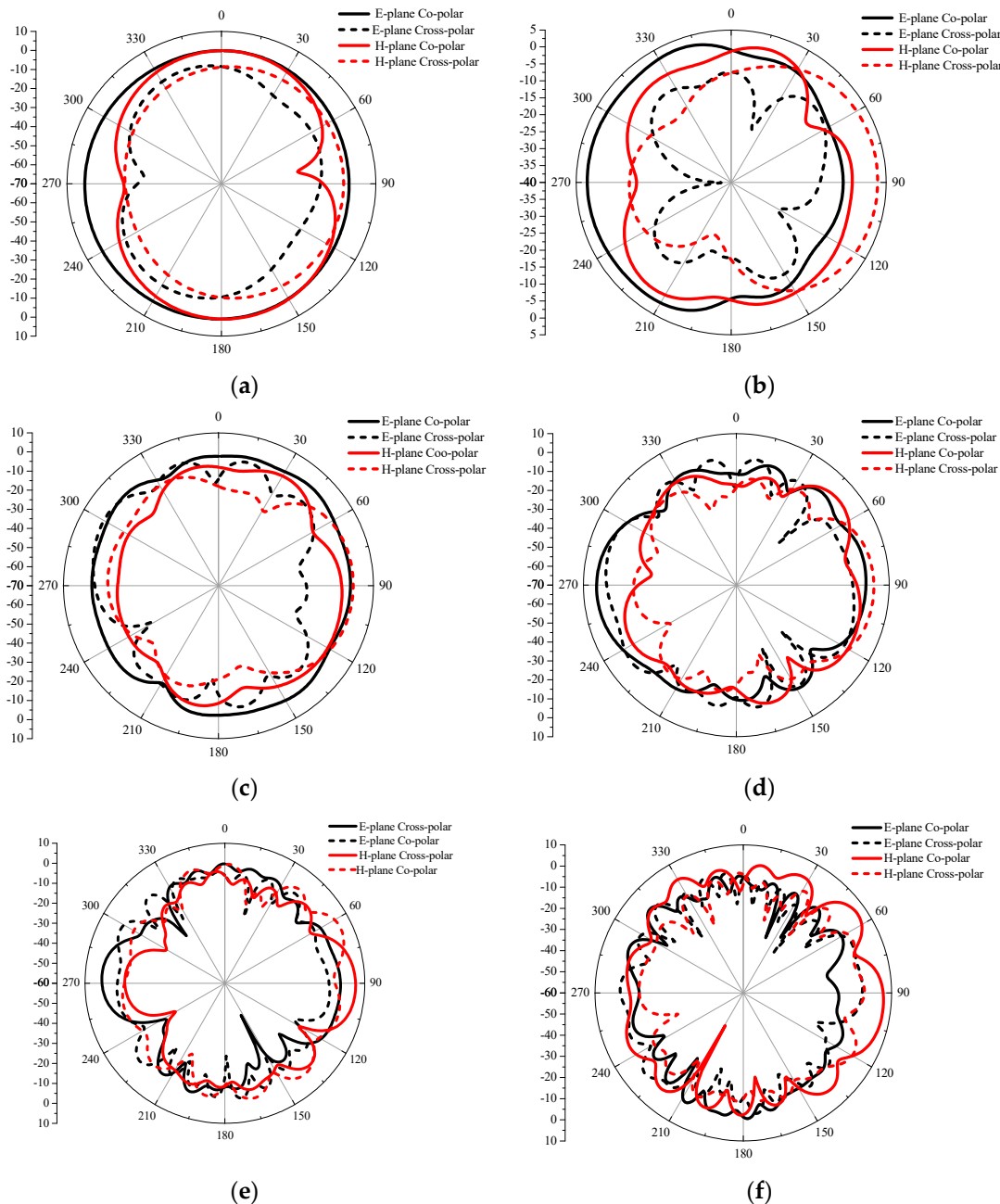

**Figure 10.** The co- and cross polar pattern in E-plane and H-plane at (**a**) 5 GHz (**b**) 10 GHz (**c**) 15 GHz (**d**) 20 GHz (**e**) 30 GHz, and (**f**) 40 GHz.

A comparison of the designed structure and previously designed structures is given in Table 1. From Table 1, it is clear that the proposed antenna has compact electrical dimensions, very large resonance frequencies with the lowest resonance frequency of 1.3 GHz, high isolation and less ECC etc. To best of our knowledge very few highly isolated MIMO antennas with lowest resonance frequency of 1.3 GHz and highest resonance frequency of 40 GHz has been designed for SWB application.

**Table 1.** Comparison to previously designed MIMO antenna.

| Ref. | Size (mm$^2$) | $f_L$–$f_H$ (GHz) | Isolation (dB) | ECC | BW Ratio | %BW | DG (dB) |
|---|---|---|---|---|---|---|---|
| [8] | 0.16λ × 0.27λ | 1.42–90 | NA | NA | 63 | 193 | NA |
| [9] | 0.37λ × 0.17λ | 1.44–18.8 | NA | NA | 13 | 172 | NA |
| [10] | 0.32λ × 0.34λ | 3.4–37.4 | NA | NA | 11 | 166.6 | NA |
| [11] | 0.45λ × 0.45λ | 1–19 | NA | NA | 19 | 180 | NA |
| [12] | 0.41λ × 0.20λ | 3–35 | NA | NA | 11.6 | 168 | NA |
| [13] | 0.33λ × 0.27λ | 1.42–90 | NA | NA | 63 | 193 | NA |
| [14] | 0.37λ × 0.23λ | 4.00–40.0 | NA | NA | 10 | 164 | NA |
| [15] | 0.31λ × 0.46λ | 3.15–32 | NA | NA | 10.1 | 164 | NA |
| [21] | 0.41λ × 0.25λ | 2.50–14.5 | <−20 | <0.04 | 5.8 | 141 | >7.4 |
| [22] | 0.42λ × 0.72λ | 3.20–10.6 | <−15 | NA | 3.3 | 107 | NA |
| [23] | 0.60λ × 0.47λ | 3.10–10.6 | <−13 | <0.02 | 3.4 | 109 | >2.5 |
| [25] | 0.51λ × 0.82λ | 3.10–10.6 | <−17 | <0.05 | 3.4 | 109 | NA |
| [27] | 0.31λ × 0.41λ | 3.10–10.6 | <−11 | <0.15 | 3.4 | 109 | NA |
| [26] | 0.33λ × 0.41λ | 2.50–11.0 | <−15 | <0.02 | 4.4 | 125 | NA |
| Proposed | 0.17λ × 0.20λ | 1.30–40.0 | <−20 | <0.02 | 30.7 | 187 | >9.99 |

## 5. Conclusions

In this paper, a compact MIMO antenna has been designed for UWB/SWB application. The semicircular shape radiator is fed with microstrip feed line. Microstrip feed line is modified to tapered feed line that provide extremely wideband resonance frequency range with lowest resonance frequency of 1.3 GHz. The proposed antenna exhibits high BW ratio, percentage BW, and has compact electrical dimension of 0.17λ × 0.20λ mm$^2$. Moreover, the mutual coupling is successfully reduced in the wideband frequency range by placing a corrugated T-shape parasitic element without affecting the impedance matching. The decoupled MIMO antenna structure has a mutual coupling less than −20 dB, a ECC less than 0.02, and a diversity gain of 10 dB.

**Author Contributions:** Writing: original draft preparation was made by H.U. (Habib Ullah), reviewed by S.U.R., I.K., H.U. (Hamid Ullah) and supervised by Q.C. All authors have read and agreed to the published version of the manuscript.

**Funding:** This work was supported by the National Natural Science Foundation of China (Grant No. 61871219).

**Acknowledgments:** The authors grateful to the College of Electronic and Information Engineering, Nanjing University of Aeronautic and Astronautics (NUAA) for providing the necessary facilities, and in particular Qunsheng Cao for supervising us.

**Conflicts of Interest:** The authors declare that there is no conflicts of interest.

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
