# Peer review of "Design of SWB MIMO Antenna with Extremely Wideband Isolation"

_electronics, doi:10.3390/electronics9010194_

Round 1
Reviewer 1 Report
The authors have proposed a small antenna with wide bandwidth for MIMO systems. The topic and concept of the proposed work are interesting, but still the current version needs to be improve before my final decision to accept it. Therefore, I encourage the authors to revise their article according with my following comments.
1) In the abstract, please provide the information about radiation properties such as gain and efficiency over the working frequency band.
2) Metamaterials and metasurfaces are very interesting techniques to reduce the mutual coupling between the radiation elements for MIMO and SAR applications. Please add a short paraghraph in the introduction section and provide explanations on these approach as well. Followings are suggested references.
"Isolation Enhancement of Densely Packed Array Antennas with Periodic MTM-Photonic Bandgap for SAR and MIMO Systems", IET Microwaves, Antennas & Propagation, online published on 18.Dec.2019, DOI: 10.1049/iet-map.2019.0362, 6 pages.
"Surface Wave Reduction in Antenna Arrays Using Metasurface Inclusion for MIMO and SAR Systems", Radio Science, 54, 1067–1075. https://doi.org/10.1029/2019RS006871.
“High-Isolation Leaky-Wave Array Antenna Based on CRLH Metamaterial Implemented on SIW with ±30o Frequency Beam-Scanning Capability at Millimeter-Waves”, Electronics, 2019, 8, 642,15 pages, doi:10.3390/electronics8060642.
"Mutual-Coupling Isolation Using Embedded Metamaterial EM Bandgap Decoupling Slab for Densely Packed Array Antennas", IEEE Access, vol. 7, pp. 5182–51840, April 29, 2019.
"Mutual Coupling Suppression Between Two Closely Placed Microstrip Patches Using EM-Bandgap Metamaterial Fractal Loading", IEEE Access, vol. 7, Page(s): 23606 – 23614, March 5, 2019.
"Interaction Between Closely Packed Array Antenna Elements Using Metasurface for Applications Such as MIMO Systems and Synthetic Aperture Radars", Radio Science, Volume53, Issue11, November 2018, Pages 1368-1381.
“Antenna Mutual Coupling Suppression Over Wideband Using Embedded Periphery Slot for Antenna Arrays”, Electronics, 2018, 7(9), 198; https://doi.org/10.3390/electronics7090198.
“Study on Isolation Improvement Between Closely Packed Patch Antenna Arrays Based on Fractal Metamaterial Electromagnetic Bandgap Structures”, IET Microwaves, Antennas & Propagation, Volume 12, Issue 14, 28 November 2018, p. 2241 – 2247.
“Meta-surface Wall Suppression of Mutual Coupling between Microstrip Patch Antenna Arrays for THz-band Applications”, Progress In Electromagnetics Research Letters, Vol. 75, page 105-111, 2018.
“A New Study to Suppress Mutual-Coupling between Waveguide Slot Array Antennas based on Metasurface Bulkhead for MIMO Systems”, Proceedings of the 2018 Asia-Pacific Microwave Conference (APMC), pp.500-502, November 6-9, 2018, Kyoto, Japan.
“New Approach to Suppress Mutual Coupling Between Longitudinal-Slotted Arrays Based on SIW Antenna Loaded with Metal-Fences Working on VHF/UHF Frequency-Bands: Study, Investigation, and Principle”, Proceedings of the 2018 Asia-Pacific Microwave Conference (APMC), pp. 1564-1566, November 6-9, 2018, Kyoto, Japan.
“Mutual-Coupling Reduction in Metamaterial SIW Slotted Antenna Arrays Using Metal Fence Isolators for SAR and MIMO Applications”, 2018 12th International Congress on Artificial Materials for Novel Wave Phenomena (Metamaterials), pp.13-15, 27.Aug. -1.Sep. 2018, Espoo, Finland.
“A New Waveguide Slot Array Antenna with High Isolation and High Antenna Bandwidth operation on Ku- and K- bands for Radar and MIMO Systems”, Proceedings of the 48th European Microwave Conference (EuMC), pp. 1421-1424, 25–27 Sep. 2018, Madrid, Spain.
“Study on Antenna Mutual Coupling Suppression Using Integrated Metasurface Isolator for SAR and MIMO Applications”, Proceedings of the 48th European Microwave Conference (EuMC), pp. 1425-1428, 25–27 Sept. 2018, Madrid, Spain.
“Mutual Coupling Reduction Using Metamaterial Supersubstrate for High Performance & Densely Packed Planar Phased Arrays”, 2018 22nd International Microwave and Radar Conference (MIKON), pp. 675 – 678, 14-17 May 2018, Warsaw Univ. of Technology, PoznaÅ„, Poland.
“Array Antenna for Synthetic Aperture Radar Operating in X and Ku-Bands: A Study to Enhance Isolation Between Radiation Elements", EUSAR 2018; 12th European Conference on Synthetic Aperture Radar, Pages: 1083-1087, 4-7 June. 2018, Aachen, Germany.
"A Technique to Suppress Mutual Coupling in Densely Packed Antenna Arrays Using Metamaterial Supersubstrate”, 12th European Conference on Antennas and Propagation (EuCAP 2018), 9-13 April 2018, London, UK.
"EM Isolation Enhancement Based on Metamaterial Concept in Antenna Array System to Support Full-Duplex Application”, 2017 IEEE Asia Pacific Microwave Conference (APMC2017), pp. 740-742, 13-16 Nov. 2017, Kuala Lumpur, Malaysia.
3) Section 2 as design section is very short, please extend it by providing more details on how to design this type of configuration? and explain why you have selected this type of layout? What are its benefits than literature?
4) Besides Fig.9 (c) please add the radiation efficiency versus frequency.
5) Table 1 as comparison table is short, please extend it by adding the above suggested works.
6) After comparison table 1, please highlight the novelty of the proposed work.
7) Is it possible to optimize the proposed array for wider frequency bands?
8) is it possible to extend the proposed array to an array with more than two radiation elements?
Author Response
Dear reviewer,
Thank you for allowing a resubmission of our manuscript, with an opportunity to address the reviewers’ comments. We would like to record our appreciation for the thoughtful, valuable and specific comments offered by the reviewers to improve the paper. We have carefully considered all their comments and have now completed the revisions incorporating all of their suggestions in the revised manuscript.
We are uploading (a) our point-by-point response to the comments (below) (response to reviewers), (b) an updated manuscript with yellow highlighting indicating changes, and (c) a clean updated manuscript without highlights (PDF main document).
|
Reviewer 1 |
|
|
Reviewer Suggestions |
Authors reply |
|
1) In the abstract, please provide the information about radiation properties such as gain and efficiency over the working frequency band. |
Gain and efficiency has been provided in the revised manuscript |
|
2) Metamaterials and metasurfaces are very interesting techniques to reduce the mutual coupling between the radiation elements for MIMO and SAR applications. Please add a short paraghraph in the introduction section and provide explanations on these approach as well. Followings are suggested references. |
We are agree with this suggestion, the suggested references are included in the revised manuscript with comprehensive explanation |
|
3) Section 2 as design section is very short, please extend it by providing more details on how to design this type of configuration? and explain why you have selected this type of layout? What are its benefits than literature?
|
In section 2 simple single antenna element is simulation to see the behavior of the unit element. Further we have mention the benefits such tapper feeding with half circular shape radiator provide wide impedance matching. |
|
4) Besides Fig.9 (c) please add the radiation efficiency versus frequency.
|
It has been added in the revised manuscript |
|
5) Table 1 as comparison table is short, please extend it by adding the above suggested works.
|
Few more references are added |
|
6) After comparison table 1, please highlight the novelty of the proposed work.
|
It has been explained on page 9 |
|
7) Is it possible to optimize the proposed array for wider frequency bands? 8) is it possible to extend the proposed array to an array with more than two radiation elements?
|
Yes we are agree with these two suggestions. We can extend the proposed design in future according to suggestion |

Reviewer 2 Report
This paper presents an antenna design for SWB MIMO. It is generally well written. The comparison has been conducted for many previous works in the literature. Please consider the followings.
In page 1, the first letter in “gigabyte” doesn’t need to be capitalized. In the last sentence of Abstract, “that” right after the comma can be removed. English needs to be reviewed throughout the paper. There are several grammatical errors.Author Response
Dear Reviewer ,
Thank you for allowing a resubmission of our manuscript, with an opportunity to address the reviewers’ comments. We would like to record our appreciation for the thoughtful, valuable and specific comments offered by the reviewers to improve the paper. We have carefully considered all their comments and have now completed the revisions incorporating all of their suggestions in the revised manuscript.
We are uploading (a) our point-by-point response to the comments (below) (response to reviewers), (b) an updated manuscript with yellow highlighting indicating changes, and (c) a clean updated manuscript without highlights (PDF main document).
|
Reviewer 2 |
|
|
In page 1, the first letter in “gigabyte” doesn’t need to be capitalized. In the last sentence of Abstract, “that” right after the comma can be removed. English needs to be reviewed throughout the paper. There are several grammatical errors. |
Yes it has been capitalized and manuscript has been reviewed to remove the grammatical errors |

Reviewer 3 Report
The paper presents a preliminary study of a proposed MIMO antenna for UWB/SWB applications. The presented results show that the proposed design is an interesting candidate for SWB applications more than for UWB applications. In my opinion it would be interesting a comparation (could be by simulation) with the sate-of-art especially the ones presented in the paper: Balani, Warsha, et al. "Design Techniques of Super-Wideband Antenna–Existing and Future Prospective." IEEE Access 7 (2019). Regarding the document format it is acceptable, but it could be improved, for example Figures should be closer to the text that mention them (figure 6 is mention on page 3 but it shows up only on page 6). Please revise the document format there are several typos.
Author Response
Dear Editor,
Thank you for allowing a resubmission of our manuscript, with an opportunity to address the reviewers’ comments. We would like to record our appreciation for the thoughtful, valuable and specific comments offered by the reviewers to improve the paper. We have carefully considered all their comments and have now completed the revisions incorporating all of their suggestions in the revised manuscript.
We are uploading (a) our point-by-point response to the comments (below) (response to reviewers), (b) an updated manuscript with yellow highlighting indicating changes, and (c) a clean updated manuscript without highlights (PDF main document).
|
Comment |
Reply |
|
The paper presents a preliminary study of a proposed MIMO antenna for UWB/SWB applications. The presented results show that the proposed design is an interesting candidate for SWB applications more than for UWB applications. In my opinion it would be interesting a comparation (could be by simulation) with the sate-of-art especially the ones presented in the paper: Balani, Warsha, et al. "Design Techniques of Super-Wideband Antenna–Existing and Future Prospective." IEEE Access 7 (2019). Regarding the document format it is acceptable, but it could be improved, for example Figures should be closer to the text that mention them (figure 6 is mention on page 3 but it shows up only on page 6). Please revise the document format there are several typos. |
Yes we are agree with is suggestion, the referenced paper is included and manuscript is modified according to the comments. The manuscript is revised to remove the grammatical errors. |
